# RegaVAE: A Retrieval-Augmented Gaussian Mixture Variational Auto-Encoder for Language Modeling

**Jingcheng Deng**[1,2], **Liang Pang**[1,*], **Huawei Shen**[1,2], **Xueqi Cheng**[1,2]

[1]Institute of Computing Technology, Chinese Academy of Sciences
[2] University of Chinese Academy of Sciences
{dengjingcheng23s, pangliang, shenhuawei, cxq}@ict.ac.cn

## Abstract

Retrieval-augmented language models show promise in addressing issues like outdated information and hallucinations in language models (LMs). However, current research faces two main problems: 1) determining what information to retrieve, and 2) effectively combining retrieved information during generation. We argue that valuable retrieved information should not only be related to the current source text but also consider the future target text, given the nature of LMs that model future tokens. Moreover, we propose that aggregation using latent variables derived from a compact latent space is more efficient than utilizing explicit raw text, which is limited by context length and susceptible to noise. Therefore, we introduce RegaVAE, a retrieval-augmented language model built upon the variational auto-encoder (VAE). It encodes the text corpus into a latent space, capturing current and future information from both source and target text. Additionally, we leverage the VAE to initialize the latent space and adopt the probabilistic form of the retrieval generation paradigm by expanding the Gaussian prior distribution into a Gaussian mixture distribution. Theoretical analysis provides an optimizable upper bound for RegaVAE. Experimental results on various datasets demonstrate significant improvements in text generation quality and hallucination removal. Our codes is released in the link[1].

## 1 Introduction

Language models (LMs) have achieved state-of-the-art performance on many NLP tasks (Zhu et al., 2021; Pang et al., 2021), which reveals that they store a large amount of world knowledge as implicit parameters. While this development is exciting, LMs still suffer from some problems (Li et al., 2022): 1) performance and model parameter size follow a power law relationship (Kaplan

---

*Corresponding Author
[1]https://github.com/TrustedLLM/RegaVAE

Table 1: Differences between RegaVAE and existing representative models.

| Model | Future Info. in | | | Aggreg-ation |
|---|---|---|---|---|
| | **Query** | **Key** | **Value** | |
| KNN-LM | ✗ | ✗ | ✔ | Explicit |
| RAG | ✗ | ✗ | ✗ | Explicit |
| REALM | ✗ | ✗ | ✗ | Explicit |
| SPALM | ✗ | ✗ | ✔ | Explicit |
| FiD | ✗ | ✗ | ✗ | Implicit |
| EMDR$^2$ | ✗ | ✗ | ✗ | Implicit |
| EPR | ✗ | ✗ | ✗ | Explicit |
| Re2G | ✗ | ✗ | ✗ | Implicit |
| RETRO | ✗ | ✗ | ✔ | Implicit |
| RegaVAE | ✔ | ✔ | ✔ | Implicit |

Table 1: Differences between RegaVAE and existing representative models. **Query**, **Key** and **Value** respectively indicate whether future information is contained in query, key and value parts. **Aggregation** represents the aggregation method of retrieved documents and source text.

et al., 2020), which results in model parameters having to grow exponentially in order to gain more world knowledge; 2) difficulty in adjusting for time-sensitive knowledge (Lewis et al., 2020); 3) may produce "fact hallucination" problem (Guu et al., 2020; Marcus, 2020).

Recently, the advent of retrieval-augmented text generation has emerged as a novel paradigm aimed at addressing these pertinent issues (Borgeaud et al., 2022; Li et al., 2022; Shi et al., 2023). Compared to generative-only models, this paradigm not only explicitly exploits similar texts to generate more fluent sentences but also leverages expertise to generate difficult responses. Nonetheless, we contend that there are two primary challenges associated with current retrieval-augmented language models. Firstly, not only current semantic information, but also future semantic information need to be considered during retrieval. Previous studies (Khandelwal et al., 2020; Guu et al., 2020; Lewis et al., 2020)

either directly use the entire text as key and value parts at the same time, and then use cosine similarity (Xu et al., 2023), TF-IDF and other indicators to search, which leads to the value part is only similar to the source text (query), and does not necessarily serve the best for generator. Another way is to divide a piece of text into two parts, where the first part and the second part are regarded as current information and future information, such as RETRO (Borgeaud et al., 2022). However, RETRO adds future information to value part, but ignores the future information in query and key, which leads to the fact that candidate documents with high similarity do not necessarily contain future information that can help the generator. Secondly, explicitly aggregating retrieved documents and source texts is limited by the length of the model input and introduces too much noise. Implicit aggregation is inefficient in irregular embedding spaces, and retrieval vectors are not generalizable.

To address the above challenges, we design RegaVAE, a **Re**trieval-augmented language model based on **ga**ussian mixture **V**ariational **A**uto-**E**ncoder. Unlike previous methods that directly encode unlabeled corpora (Karpukhin et al., 2020; Lewis et al., 2020) or only adding future information to the value part (Borgeaud et al., 2022), as shown in Tab. 1, our model considers future information through a latent space, given an x, we decode it into a y using a conditional VAE, which ensures that the latent variables contain information from both source and target data. In addition, in order to implicitly aggregate the retrieved documents and source texts, we also use the probabilistic form of the retrieval generation paradigm to theoretically extend the prior Gaussian distribution to a Gaussian mixture distribution. This allows the latent space to satisfy continuity and uniformity, and the latent vector after aggregating retrieved documents and source text has better representation ability. Tab. 1 summarizes the differences between RegaVAE and existing representative methods. Overall, our contributions are as follows:

- We propose a retrieval method that implicitly combines current and future information, which introduces future information into the query, key, and value parts at the same time, so that the higher the document similarity, the more helpful it is for the generator.

- We integrate the VAE and retrieval generation probabilistic framework to efficiently aggregate retrieval information into the generation process. Furthermore, we derive an upper bound on the optimization of this framework.

- Experiments have shown that RegaVAE is competitive in generating quality, generating diversity, and eliminating hallucinations.

## 2 Related Work

We classify related studies into two categories, explicit aggregation and implicit aggregation, according to the way the retrieved documents and source text are aggregated. Explicit aggregation refers to concatenating retrieved documents directly into source text to construct augmented input. Implicit aggregation refers to adding retrieved documents to the generator in the form of vectors or distributions.

**Explicit Aggregation** Guu et al. (2020) proposed an end-to-end framework REALM that achieves state-of-the-art performance on three open-domain QA. A representative work is RAG (Lewis et al., 2020), which first uses DPR (Karpukhin et al., 2020) to retrieve relevant documents, and then links relevant documents with the source text for sequence-to-sequence generation. Different from RAG and REALM, Rubin et al. (2022) proposed EPR, which is a method for retrieving prompts and can improve the effect of prompts. Re2G (Glass et al., 2022) is an enhanced version of RAG, which improves the quality of retrieved documents by integrating multiple retrieval methods. Explicit aggregation is simple and effective, but it suffers from the limitation of the input length of the language model and cannot fully utilize the large number of retrieved documents. In addition, it is easy to introduce noise, making the model performance unstable. Unlike these methods, our model implicitly aggregates retrieved documents into the generation process.

**Implicit Aggregation** FiD (Izacard and Grave, 2021) uses a DPR to retrieve candidate documents, and then splices and encodes the candidate documents with the source text, and inputs them into the generator in the form of vectors. EMDR$^2$ (Sachan et al., 2021) is similar to FiD, and it provides an end-to-end framework to train both the retriever and the generator. However, the query, key and value parts of FiD and EMDR$^2$ do not contain future information, which will cause the value part to be similar to the query part and not conducive to the generation of future tokens. RETRO (Borgeaud

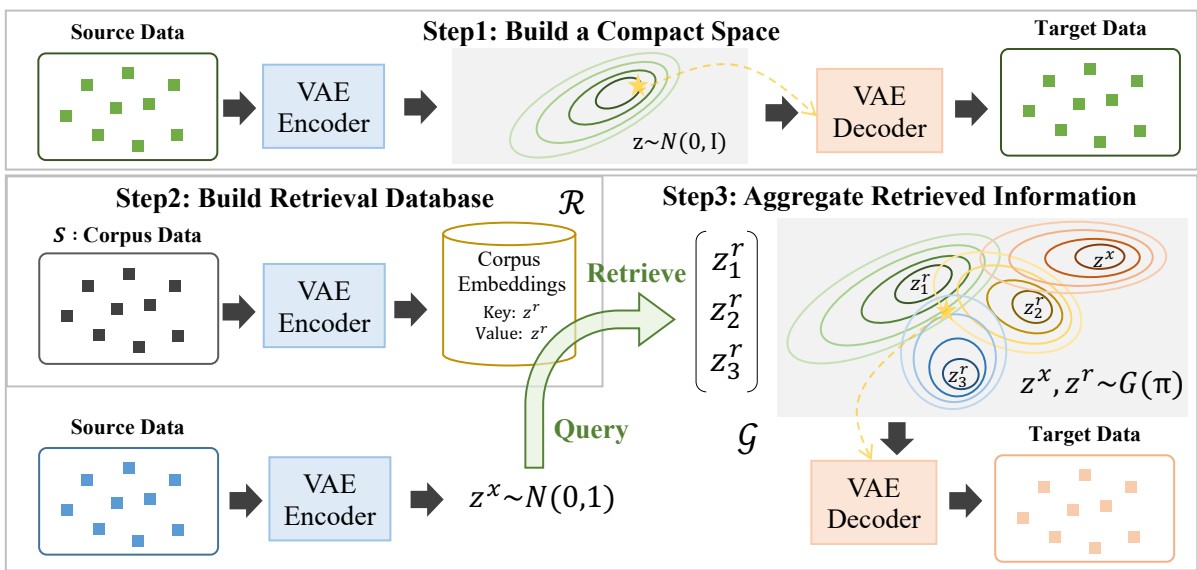

Figure 1: Architecture of RegaVAE. Based on the training data, we first train a VAE to construct a compact latent space, which ensures that the latent variable $z$ contains both current and future information (see § 3.1). We then build a retrieval database and then aggregate the retrieved information into the generator (see § 3.2). VAE Encoder and Decoder parameters are the same in all steps. In order to ensure fairness, the Corpus data and the Source data in the training set are the same. $G$ represents the Gaussian mixture distribution, and $\pi$ is the corresponding parameter.

et al., 2022) and KNN-LM (Khandelwal et al., 2020) set key and value parts as a piece of text, and added the continuation and the next token of this text in value part, respectively. However, they only calculate the similarity between the query and key while ignoring future information in value part, resulting in high similarity documents containing future information that may not necessarily help the generator. Our model sets both key and value parts as latent variables of a piece of text and its future continuation, and the query encoded by the VAE encoder also contains future information, so future information is also taken into account when calculating the similarity between query and key, making up for the shortcomings of previous studies.

## 3 Methodology

Most text generation tasks can be formulated as a mapping from a source text $x$ to a target text $y : y = f(x)$, while retrieval-augmented text generation can be further formulated as: $y = f(x, r)$, where $r$ is the relevant document retrieved based on $x$. Specifically, this approach generally encompasses the utilization of a retriever denoted as $\mathcal{R}$ and a generator denoted as $\mathcal{G}$. The retriever $\mathcal{R}$ obtains $r$ from the retrieval source $\mathcal{S}$ by the retrieval metric $D$ and $x$. Then $r$ and $x$ are fed into $\mathcal{G}$ to obtain $y$ through a predefined integration method $I$.

Commonly used retrieval indexes $D$ include cosine similarity, TF-IDF, etc. This paradigm can also be expressed in probabilistic form:

$$p(y|x) = \sum_{r \in \text{top-}k(p(\cdot|x))} p(y|x, r)p(r|x). \quad (1)$$

Next, the framework of RegaVAE is introduced, which consists of three steps. Firstly, in order to construct a compact space, we introduce the VAE structure. Since transformers based on VAE all suffer from the posterior collapse (Fu et al., 2019), we follow a previous study (Hu et al., 2022) which combines low-rank tensor products for latent variables and decoders (see § 3.1 and step 1 in Fig. 1). Secondly, to introduce retrieval information into the latent space, we first introduce how the retrieval library is constructed (see step 2 in Fig. 1), and then replace the prior Gaussian distribution in the original VAE with a Gaussian mixture distribution to derive RegaVAE (see step 3 in Fig. 1). This allows for deep aggregation of retrieved and input documents and simultaneously incorporates future information into query, key and value parts, which helps to generate more fluent sentences (see § 3.2) . Finally, to train RegaVAE, we derive an optimizable upper bound on the loss function for unclosed solutions (see § 3.3). Fig. 1 shows the whole framework diagram.

### 3.1 Introduce Retrieval Information into Latent Space

We consider using the VAE structure to make the space compact and continuous. As a kind of generative model, VAE estimates the intractable data distribution $p(x)$ by deriving and maximizing its **E**vidence **L**ower **BO**und (ELBO) as:

$$
\begin{aligned}
\log p(x) \geq \mathcal{L}_{\text{ELBO}} = \\
\mathbb{E}_{q_\phi(z|x)}[\log p_\theta(x|z)] - \text{KL}(q_\phi(z|x)||p(z)),
\end{aligned}
\tag{2}
$$

where $z$ is the latent variable. $p(z)$ and $p(z|x)$ is the prior and posterior distribution of $z$, respectively. $q_\phi(z|x)$ and $p_\theta(x|z)$ represent Encoder and Decoder. $\theta$ and $\phi$ are corresponding parameters.

Due to the power of the decoder, transformers based on VAE usually have the problem of posterior collapse. According to Hu et al. (2022), we use a low-rank tensor product in the $l$-th layer of the model:

$$
\tilde{v}_i^{(l)} = (\sum_{j=1}^r W_v^{(l,j)} v_i^{(l)}) \circ (\sum_{j=1}^r W_z^{(l,j)} z_l),
\tag{3}
$$

where $z_l$ and $v^{(l)}$ represent latent variable and hidden variable of $l$-th layer respectively. $v_i^{(l)}$ represents the hidden vector of the $i$-th token in $l$-th layer. $r$ is a hyper-parameter, and $\circ$ means element-wise multiplication. $W_v$ and $W_z$ are learnable parameters which are shared across all positions ($i$) but not shared with $l$-th layer.

In order not to introduce additional data, we use the training set as the data for training VAE. By optimizing ELBO, each sample is encoded into the latent space and then restored by the decoder to obtain a compact latent space.

### 3.2 Build the RegaVAE Model

**Build Retrieval Database** With a compact latent space, we use an encoder to encode $x$ and $r$ from $\mathcal{S}$ into the latent space. The latent variables of $x$ and $r$ are denoted by $z^x$ and $z^r$, respectively. Then we store $z^r$ as key and value parts in the retrieval database. Given a query $z^x$, we compute the inner product of it and $z^r$ to obtain the similarity.

$$
D(z^x, z_i^r) = \cos(z^x, z_i^r),
\tag{4}
$$

where $z_i^r \sim N(\mu_i, \sigma_i^2)$ represents the latent vector of the $i$-th retrieved sample in $\mathcal{S}$. $\mu_i$ and $\sigma_i^2$ are the corresponding mean and standard deviation, respectively. Since our framework is trained end-to-end, the parameters of the encoder change with

each training step, resulting in changes in the latent space. Considering that it is impractical to update the retrieval database in real-time, and previous work (Guu et al., 2020) has shown the practicality of updating the index intermittently during training, we follow this approach and update the index of retrieval database every fixed number of training steps.

**Aggregate Retrieved Information** Inspired by the retrieval-generated text generation paradigm, we assume $y$ is influenced by latent variables $z^x$ and $z^r$. To obtain the ELBO of RegaVAE, we first model $\log p(y)$ as:

$$
\begin{aligned}
\log p(y) &= \log \iint p(y, z^r, z^x) dz^r dz^x \\
&\geq \iint \log p(y, z^r, z^x) dz^r dz^x \\
&= \iint \log q(z^r, z^x|x) \frac{\log p(y, z^r, z^x)}{\log q(z^r, z^x|x)} dz^r dz^x \\
&= \mathbb{E}_{q(z^r, z^x|x)} \log[\frac{p(y, z^r, z^x)}{q(z^r, z^x|x)}].
\end{aligned}
\tag{5}
$$

From the first step, the Jensen inequality can be used to transform to the second step, and then the expression of the desired form can be obtained. According to Bayes formula:

$$
p(y, z^r, z^x) = p(y|z^r, z^x)p(z^r, z^x).
\tag{6}
$$

Substituting Eq. 6 into Eq. 5:

$$
\begin{aligned}
&\mathbb{E}_{q(z^r, z^x|x)} \log[\frac{p(y, z^r, z^x)}{q(z^r, z^x|x)}] \\
&= \mathbb{E}_{q(z^r, z^x|x)} \log[\frac{p(y|z^r, z^x)p(z^r, z^x)}{q(z^r, z^x|x)}] \\
&= \mathbb{E}_{q(z^r, z^x|y)}[\log p(y|z^r, z^x)] \\
&\quad - \text{KL}(q(z^r, z^x|x)||p(z^r, z^x)),
\end{aligned}
\tag{7}
$$

where KL stands for calculating the KL divergence between two distributions. Eq. 7 is the ELBO of RegaVAE. At this point, Eq. 7 and Eq. 2 have the same form, but the latent variable $z$ is replaced by $z^x$ and $z^r$. Since each $z_i^r$ follows a Gaussian distribution, we consider using a Gaussian mixture distribution to combine $z^x$ and $z^r$. So $q(z^r, z^x|x)$ can be expressed as:

$$
q(z^r, z^x|x) = w_0 q(z^x|x) + \sum_{i=1}^n w_i q(z_i^r|x),
\tag{8}
$$

where $n$ represents the number of retrieved documents.

$$
w_i = \text{softmax}(D(z^x, z_i^r)),
\tag{9}
$$

where $\sum_{i=0}^{n} w_i = 1$ makes $q(z^r, z^x|x)$ satisfy the requirement of Gaussian mixture distribution. So far, we have obtained the theoretical framework for introducing retrieval information in latent space.

### 3.3 Training RegaVAE

We can optimize RegaVAE by optimizing Eq. 7. In the KL divergence term of Eq. 7, the closed-form solution cannot be obtained because the two distributions are mixed Gaussian distributions. Therefore, we continue to use previous research (Dilokthanakul et al., 2016), that is, to optimize its upper bound. First we assume two Gaussian mixture distributions as:

$$p = \sum_{i=1}^{n} \pi_i g_i, \quad \hat{p} = \sum_{i=1}^{n} \hat{\pi}_i \hat{g}. \quad (10)$$

The KL divergence between them can be expressed as:

$$\begin{aligned}
\mathrm{KL}(p||\hat{p}) &= \int \left( \sum_i \pi_i g_i \right) \log \frac{\sum_i \pi_i g_i}{\sum_i \hat{\pi}_i \hat{g}_i} \\
&\leq \int \sum_i \pi_i g_i \log \frac{\pi_i g_i}{\hat{\pi}_i \hat{g}_i} \\
&= \sum_i \pi_i \log \frac{\pi_i}{\hat{\pi}_i} + \sum_i \pi_i \int g_i \log \frac{g_i}{\hat{g}_i} \\
&= \mathrm{KL}(\pi||\hat{\pi}) + \sum_i \pi_i \mathrm{KL}(g_i||\hat{g}_i).
\end{aligned} \quad (11)$$

In the variational distribution $q(z^r, z^x|x)$, the trainable parameter is only $w_i$ and $q(z^x|x)$. And the prior distribution $p(z^r, z^x)$ is defined as:

$$p(z^r, z^x) = \hat{w}_0 p(z^x) + \sum_{i=1}^{n} \hat{w}_i p(z_i^r), \quad (12)$$

where $z^x \sim N(0,1)$ and $z_i^r \sim N(0,1)$. So the upper bound for the KL term in Eq. 7 can become:

$$\begin{aligned}
\mathrm{KL}(q(z^r, z^x|x)||p(z^r, z^x)) &\leq \sum_{i=0}^{n} \mathrm{KL}(w_i||\hat{w}_i) \\
+ \mathrm{KL}(q(z^x|x)||N(0,I)) + \mathrm{C},
\end{aligned} \quad (13)$$

where C is a constant that has nothing to do with model parameter updates. We do not update the retrieval library in real time, but regularly update it according to the number of training steps. In this setup, $w_i$ is constant, so Eq. 13 becomes:

$$\begin{aligned}
\mathrm{KL}(q(z^r, z^x|x)||p(z^r, z^x)) \\
\leq \mathrm{KL}(q(z^x|x)||N(0,1)) + \mathrm{C}.
\end{aligned} \quad (14)$$

Substituting Eq 14 into Eq 7, we can get the final optimized loss function:

$$\begin{aligned}
\mathcal{L} = \mathbb{E}_{q(z^r, z^x|y)}[\log p(y|z^r, z^x)] \\
- \mathrm{KL}(q(z^x|x)||N(0,1)).
\end{aligned} \quad (15)$$

Eq. 15 can be regarded as an optimizable upper bound of Eq. 8. When given a dataset, we first encode the source text to obtain a retrieval database. The top-k documents are then retrieved for each $x$ separately. Then the corresponding latent variables $z^x$ and $z^r$ are aggregated in the form of Gaussian mixture distribution and then input into $\mathcal{G}$ to obtain the output. Finally, we use Eq. 15 to train RegaVAE.

## 4 Experiment

This section provides the experimental datasets, experimental settings, and experimental results.

### 4.1 Datasets

For experiments, we employ three datasets, namely Yelp (Yang et al., 2017), Yahoo (He et al., 2019) and WritingPrompts (WP) (Fan et al., 2018). As in previous studies (Hu et al., 2022), due to the limitation of computing resources, we adopt the methodology established in previous research and sample 100,000 data instances from the training set of Yelp and Yahoo for model training. This consistent approach ensures a fair and equitable basis for comparison across the evaluated models.

### 4.2 Metrics

**Generation Quality** In the context of the text generation task, we present the evaluation metrics of perplexity (PPL), Self-BLEU (Zhu et al., 2018), Dist2 (Li et al., 2016), and Activation Units (AU) (Burda et al., 2016). For the WritingPrompts, in addition to PPL, we also report the metrics of BLEU (Papineni et al., 2002), Rouge-1, Rouge-2, Rouge-L (Mithun et al., 2012), and BERTScore (Zhang et al., 2020).

**Hallucination** We use SelfCheckGPT (Manakul et al., 2023) to detect hallucinations produced by the model. There are four indicators in total, namely $\mathrm{S_{BERT}}$, $\mathrm{S_{QA}}$, $\mathrm{S_n^a}$ and $\mathrm{S_n^m}$. The higher their value, the more likely the model is hallucinating.

### 4.3 Experiment Settings

We have chosen two distinct categories of models as our baselines. The first category comprises transformers based on VAE, and the second category consists of retrieval-generated models. These baselines provide us with a comprehensive framework for evaluating and contrasting different approaches.

| Model | Yelp | | | | Yahoo | | | | Cost |
|---|---|---|---|---|---|---|---|---|---|
| | PPL↓ | Self-BLEU↓ | Dist2↑ | AU↑ | PPL↓ | Self-BLEU↓ | Dist2↑ | AU↑ | |
| GPT2 | 22.13 | 65.90 | 17.96 | - | 24.17 | 54.06 | 21.07 | - | - |
| *Retrieval-augmented Language Model* | | | | | | | | | |
| KNN-LM | 39.95 | - | - | - | 62.30 | - | - | - | 8 |
| FiD | 14.08 | 42.26 | 24.45 | - | 14.71 | 42.84 | 26.49 | - | 66 |
| RETRO | 16.53 | 46.65 | 23.23 | - | 13.27 | 38.64 | 28.83 | - | 44 |
| RAG | 20.68 | 58.53 | 28.16 | - | 17.62 | 48.91 | 24.95 | - | 58 |
| *Transformers based on VAE* | | | | | | | | | |
| Optimus | 22.79 | - | - | - | 23.11 | - | - | - | - |
| Embed | 19.98 | 65.27 | 15.59 | 6 | 22.18 | 54.15 | 20.80 | 3 | - |
| Memory | 19.95 | 63.90 | 16.91 | 11 | 22.03 | 54.59 | 21.87 | 18 | - |
| Softmax | 20.14 | 64.26 | 16.51 | 13 | 22.35 | 54.49 | 21.65 | 19 | - |
| ADAVAE | 15.49 | 49.80 | - | 32 | 14.23 | - | - | 32 | - |
| DELLA | 12.35 | 60.02 | 17.63 | 23 | 11.49 | 48.53 | 21.88 | 21 | - |
| RegaVAE | **8.62** | **36.10** | **28.83** | **52** | **6.99** | **30.74** | **33.03** | **56** | 60 |

Table 2: Results for the Yelp and Yahoo. For transformers based on VAE, results of Optimus are directly copied from the original paper with $\lambda = 0.5$. The activation threshold of AU is 0.2. For retrieval-augmented language models, RETRO, FiD and RAG are reproduced by ourselves under the same parameter size. KNN-LM employs the training set data as the retrieval corpus. In addition, to ensure fairness, all retrieval sources are training sets. The Cost column provides an indication of the temporal investment(h) required for training the respective model on an A40-48G GPU.

**Transformers based on VAE** For a comprehensive comparison, we choose Optimus (Li et al., 2020) and ADAVAE (Tu et al., 2022) as the baseline models, along with four distinct paradigms: Embed (Li et al., 2020), Memory (Fang et al., 2021), Softmax (Wang and Wan, 2019) and DELLA (Hu et al., 2022). Optimus is a large-scale model based on VAE that utilizes a pre-trained BERT model as its encoder and a pre-trained GPT-2 model as its decoder. In order to ensure the fairness of the evaluation, RegaVAE uses the same pre-trained language model as Embed, Memory, Softmax and DELLA. This selection facilitates a rigorous and unbiased comparative analysis across these models.

**Retrieval-augmented Language Model** According to the division method of related work, we select representative works from different categories of retrieval-augmented language models as baselines. Specifically, RAG, FiD, and RETRO represent models with explicit aggregation, implicit aggregation without future information, and implicit aggregation with only future information in value part, respectively.

**Our Model** Consistent with prior research, we adopt the GPT2 model as the underlying backbone network for our experimentation. The dimension of

the hidden variable is set to 32, and KL annealing (Fu et al., 2019) is implemented to mitigate the issue of KL term disappearance. The learning rate is fixed at $5 \times 10^{-5}$ to ensure stable training. Our training procedure entails an initial 10 epoch training phase on the original DELLA model to establish a robust initial VAE space. Subsequently, we conduct approximately fifteen epochs of training on the RegaVAE model until it achieves convergence. To make the training process more efficient, we precomputed document embeddings for the training dataset and created a FAISS index (Johnson et al., 2021) for fast similarity searches. We use the bert_score library [2] to calculate the BERTScore for our models and baselines.

### 4.4 Automatic Evaluation

**Text Generation** Tab. 2 presents the results attained by RegaVAE model on text generation datasets. Compared to the three baseline models for retrieval augmentation, our model achieves substantial improvements in all metrics, and performs particularly well in generating quality metrics. The enhanced PPL, Self-BLEU, and Dist2 scores demonstrate that latent variables, which contain both source and target information, combined

---

[2] https://github.com/Tiiiger/bert_score

| Model | PPL↓ | BLEU↑ | R1↑ | R2↑ | RL↑ | BERTScore↑ | Self-BLEU↓ | Dist2↑ |
|---|---|---|---|---|---|---|---|---|
| GPT2 | - | 27.89 | 27.72 | 7.96 | 14.30 | 78.12 | 53.78 | 22.99 |
| Embed | - | 39.67 | **36.17** | 7.96 | 15.78 | 81.64 | 64.55 | 14.31 |
| Memory | - | 40.79 | 36.13 | 8.04 | 16.16 | 81.68 | 67.56 | 12.90 |
| Softmax | - | 41.04 | 36.14 | 8.12 | 16.30 | 81.75 | 67.02 | 13.08 |
| DELLA | 2.16 | 41.39 | 35.46 | 8.78 | 17.20 | 81.77 | 56.28 | 20.91 |
| RegaVAE | **1.18** | **43.83** | 32.21 | **9.62** | **30.57** | **84.31** | **52.70** | **23.28** |

Table 3: Results for the WritingPrompts. R1, R2 and RL represent Rouge-1, Rouge-2 and Rouge-L, respectively. The results for GPT2, EMbed, Memory and softmax are from the DELLA paper.

| Model | $S_{BERT}$↓ | $S_{QA}$↓ | $S_n^a$↓ | $S_n^m$↓ |
|---|---|---|---|---|
| FiD | 8.27 | 37.99 | 4.96 | 6.31 |
| RETRO | **7.94** | 39.84 | 4.89 | 5.78 |
| RAG | 8.52 | 38.78 | 5.04 | 5.76 |
| DELLA | 8.41 | 40.01 | 5.21 | 5.30 |
| RegaVAE | 8.01 | **37.82** | **4.42** | **4.89** |

Table 4: Hallucination evaluation results on the Yelp dataset.

| Model | Flu.↑ | Coh.↑ | Div.↑ | Hal.↑ |
|---|---|---|---|---|
| FiD | 3.64 | 2.96 | 3.17 | 3.83 |
| RETRO | 3.33 | 3.11 | 3.32 | 4.01 |
| RAG | 3.15 | 2.73 | 3.25 | 3.99 |
| DELLA | 3.67 | **3.31** | 3.15 | 3.90 |
| RegaVAE | **3.78** | 3.21 | **3.47** | **4.11** |

Table 5: Human evaluation results on the Yelp dataset.

with the extension to Gaussian mixture priors, effectively enhances the fluency and diversity of the generated text. This empirical validation corroborates the theoretical strength of our model.

Notably, in comparison to the transformer-based VAE model, RegaVAE with retrieval demonstrates superior performance in terms of both generative diversity and quality. This enhancement can be attributed to the utilization of a Gaussian mixture distribution, which offers the ability to capture multimodal distributions more effectively than a single Gaussian distribution. Leveraging the source data, RegaVAE retrieves auxiliary latent variables that facilitate the generation of the target data, thereby yielding improved text generation outcomes. Furthermore, the significant improvement in the AU value indicates that the aggregation we employ positively contributes to the generative process of decoder. This alleviates the problem of collapsing at the rear of the model to a considerable extent.

**Hallucination Evaluation** We evaluate hallucinations of RegaVAE on the Yelp dataset. Specifically, we sample the text generated by the same latent variable three times, and then feed the sampling results into SelfCheckGPT to obtain evaluation scores. The results are shown in the Tab. 4. From the experimental results, it can be seen that the text generated by RegaVAE is the least hallucinatory compared with other models.

## 4.5 Human Evaluation

In addition to automated evaluation, we conducted a human evaluation to assess and compare the performance of baseline models against our proposed method. Five professionals with expertise in the domain were enlisted to participate in the manual evaluation process. Each evaluator was tasked with rating the attributes of fluency (Flu.), coherence (Coh.), diversity (Div.), and hallucination (Hal.) on a scale ranging from 1 to 5. A rating of 1 denoted very low performance, while a rating of 5 indicated very high performance. A total of 50 test samples were randomly selected and evaluated across different models. The final human evaluation result was obtained by averaging the scores provided by the evaluators.

Tab. 5 presents the outcomes of human evaluation conducted on the Yelp dataset. RegaVAE outperforms the baseline models in almost all dimensions, demonstrating superior performance in comparison. To further establish the correlation between human and automated evaluation results, we calculated the Pearson correlation coefficient and presented the corresponding values in Tab. 6. The results obtained from human evaluation align closely with those derived from partially automated evaluation metrics. For example, the correlation between the human evaluation metrics (Flu., Coh.) associated with PPL and PPL itself is nearly identical.

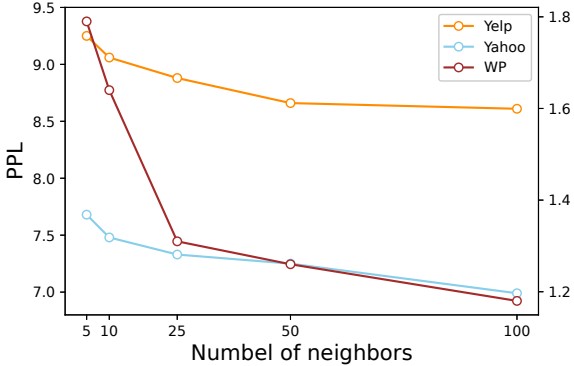

Figure 2: Performance of RegaVAE on test sets as a function of the number of retrieved neighbors. The brown broken line corresponds to the scale on the right.

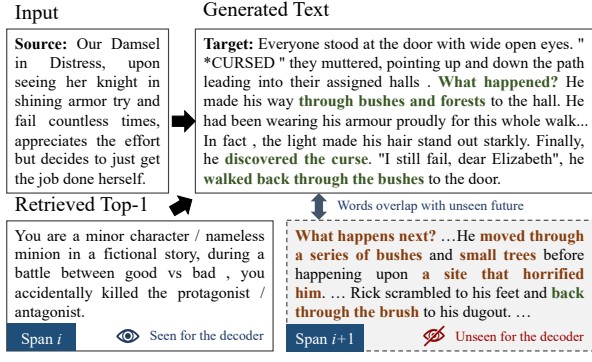

Figure 3: Generation examples of RegaVAE from test set of WritingPrompts. Green font and brown font represent the future information of the retrieved document and the relevant text generated by RegaVAE, respectively.

## 5 Analysis

To further analyze RegaVAE, we explore the impact of the number of retrieved neighbors, different model structures on the model performance. We also give a case to verify the model performance.

### 5.1 Number of Retrieved Neighbors

Fig .2 depicts the performance trends in relation to the number of retrieved neighbors. Notably, as the number of retrieved neighbors increases from 5 to 100, we observed a reduction in PPL by 0.64 on the Yelp dataset and 0.69 on the Yahoo dataset, and PPL on the WP dataset is reduced by 0.59. This upward trend proves that implicit aggregation methods can effectively filter noise compared to explicit aggregation methods, and moreover, aggregations using Gaussian mixture distributions are effective for retrieving documents and source texts.

| $\mathbf{Cor_{p\text{-value}}}$ | Flu. | Coh. | Div. | Hal. |
|---|---|---|---|---|
| PPL | $94_{0.02}$ | $86_{0.07}$ | $40_{0.51}$ | $26_{0.68}$ |
| Self-BLEU | $51_{0.37}$ | $19_{0.76}$ | $66_{0.23}$ | $35_{0.57}$ |
| Dist2 | $22_{0.73}$ | $57_{0.31}$ | $67_{0.21}$ | $58_{0.30}$ |

Table 6: Correlation between human and automated evaluation results. The order in which the model results were used to calculate the correlation coefficient is: FiD, RETRO, RAG, DELLA, and RegaVAE. The correlation coefficients have been processed with absolute value and amplified by a factor of one hundred from their original values.

| Model | PPL↓ | Self-BLEU↓ | Dist2↑ |
|---|---|---|---|
| Ours | 8.62 | 36.10 | 28.83 |
| *Aggregation Method* | | | |
| w/o VAE & R | 17.95 | 53.24 | 16.46 |
| w/o VAE | 20.13 | 62.48 | 17.04 |
| *Retrieval Method* | | | |
| base+BM25 | 13.49 | 55.37 | 20.72 |
| base+DPR | 12.58 | 58.46 | 20.54 |

Table 7: Results of ablation experiments on the Yelp dataset. **w/o VAE** represents the removal of VAE space, and **w/o VAE & R** represents the removal of VAE space and retrieval operations. **base** represents the RegaVAE that removes the retrieval.

### 5.2 Ablation Experiment

To evaluate the effectiveness of the model structure, we conducted ablation experiments involving retrieval and aggregation, as depicted in Tab. 7. When we excluded the VAE structure, there was a notable decline in the performance of RegaVAE. Interestingly, we observed that the model augmented with retrieval performed even worse than the model without retrieval when the VAE structure was absent. We speculate that the retrieved variables in this particular scenario reside in a space that fails to meet the requirements of uniformity and continuity. As a result, the model struggled to generate valid samples based on cosine similarity, introducing unwanted noise instead.

Compared with other retrieval methods, it can be seen that the performance of traditional retrieval methods is obviously insufficient. This discrepancy can be attributed to our approach incorporating future information into key, value, and query parts simultaneously, thus taking future information into account in both retrieval and generation phases, further validating our motivation.

## 5.3 Case Study

We present a compelling example to examine the quality of RegaVAE-generated text and explore the integration of retrieval information into the generated content, as illustrated in Fig. 3.

Through our observations, we have noted that the text produced by RegaVAE demonstrates a remarkable ability to establish a coherent connection with the source text while being vivid and specific. Moreover, despite encoding only the retrieved document into the latent space and subsequently integrating it into the generation process, it is evident that RegaVAE-generated text effectively incorporates future information from the retrieved document.

## 6 Conclusion

In this paper, we summarize two major challenges of existing retrieval-augmented language model methods, and propose RegaVAE to address them. We find that RegaVAE outperforms traditional retrieval generative models in terms of both generative quality and reduce hallucinations. In addition, ablation experiments and three analysis experiments verify the correctness of the model motivation. In future work, we will consider migrating RegaVAE to large language models.

## Limitations

At present, almost all large language models are pre-trained on large-scale corpus, and due to the limitation of computing resources, we cannot pretrain RegaVAE on large-scale corpus, which will lead to performance degradation.

Furthermore, the model is not stable to train due to the posterior collapse problem. Even if we adopt a low-rank tensor product, this problem still cannot be completely solved.

## Ethics Statement

We honor and support the EMNLP code of Ethics. This paper mainly studies the use of retrieval generation to eliminate the illusion in the language model and make the generated text more fluent. Our method can introduce canonical text to make language models more reliable. In addition, the data sets used in this article are all open source and do not involve any privacy or ethical issues.

## Acknowledgement

This work was supported by the National Key R&D Program of China (2022YFB3103700, 2022YFB3103704), the National Natural Science Foundation of China (NSFC) under Grants No. 62276248, and the Youth Innovation Promotion Association CAS under Grants No. 2023111.

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
