# OpenReview forum: "RegaVAE: A Retrieval-Augmented Gaussian Mixture Variational Auto-Encoder for Language Modeling"
_EMNLP/2023/Conference — EMNLP 2023 Findings_

### Official Review · Reviewer_RR3J · 2023-07-25

**Soundness:** 2

**Excitement:**

2: Mediocre: This paper makes marginal contributions (vs non-contemporaneous work), so I would rather not see it in the conference.

**Paper Topic And Main Contributions:**

This paper focuses on issues of current research and proposes RegaVAE, a retrieval-augmented language model built upon the variational auto-encoder (VAE).
Authors argue that valuable retrieved information should not only be related to the current source text but also consider the future target text.
That is, RegaVAE encodes the text corpus into a latent space, capturing current and future information from both source and target text.
Then, it initializes the latent space and adopts the probabilistic form of the retrieval generation paradigm by expanding the Gaussian prior distribution into a Gaussian mixture distribution.
Its contribution is to outperform traditional retrieval generative models in terms of both generative quality and reduce hallucinations.

**Questions For The Authors:**

In 3rd line of Eq (5), $\ge$ is right?

Are $g_{i}$ and $\hat{g}_{i}$ Gaussian mixture?

Is $\int \hat{g}_{i}=1$ right?

**Reasons To Accept:**

* The issues raised are understandable
  * The valuable retrieved information should not only be related to the current source text but also consider the future target text.
* Challenges associated with current retrieval-augmented language models are also interesting
  * RegaVAE considers future information through a latent space, given an $x$, it is decoded into a $y$ using a conditional VAE, which
    ensures that the latent variables contain information from both source and target data.
* Limitations and Ethics Statements are good

**Reasons To Reject:**

* Doubts remain about the appropriateness of this approach to solving the problem of retrieval-augmented language models.
  * In-context learning with LLMs appears to be a promising approach but has not been fully explored. Since the only comparison being made is Rubin et al. (2022), it cannot be said that RegaVAE implicitly aggregates retrieved documents into the generation process (L.134-147).
  * Not only that, but the existing research on VAE-based generation models are weak.
* Weak experimentation
  * Baseline models are inadequate and outdated. This comparison does not adequately demonstrate superiority.
* The writing in this paper is lacking in substance and poorly organized

**Reproducibility:**

3: Could reproduce the results with some difficulty. The settings of parameters are underspecified or subjectively determined; the training/evaluation data are not widely available.

**Reviewer Confidence:**

5: Positive that my evaluation is correct. I read the paper very carefully and I am very familiar with related work.

---

> ### Author Rebuttal · Authors · 2023-08-28
>
> We thank the reviewer for the detailed comments. In the following, we explain each of your concerns.
>
> **Q:** *Reason 1*
>
> **A:** In our definition, explicit aggregation means inserting the retrieved documents as prefixes before the original samples, which are then input to the decoder for training (or inference). The implicit aggregation is defined as turning the original sample and the retrieved documents into embedding vectors, merging them at the vector level and then handing them over to the decoder for training (or inference) (this is exactly what RegaVAE uses). In lines 139-147, we state the disadvantages of display aggregation (some additional references are cited to demonstrate our point). In Sec 5.1 we proved that implicit aggregation has the advantage of filtering irrelevant document information from the perspective of retrieved neighbors. In addition, In-context Learning is for LLMs. Retrieval-based In-context Learning is to input retrieved documents as context and questions into LLM for inference and does not involve the training process. For comparison with this method, we use the RAG baseline model, which differs from In-context Learning only by having an additional training process. It is proved in Tab. 2 that RegaVAE is better than RAG on the same scale.
> Secondly, existing VAE-based models do not have as many parameters as RegaVAE. Although VAE-based models are not as popular as LLM today, it is still a good research area because of strong interpretability and regular latent space. It can be seen from Tab. 2 that under the same parameter scale, the retrieval generation model with VAE (RegaVAE) is better than the retrieval generation model without VAE. In addition, it is also feasible to extend RegaVAE to LLM. One way is to implement the RegaVAE framework by adding a small number of fine-tunable parameters to the original LLM. This allows effective fine-tuning of parameters and reduces the cost of application on LLM. If we implement this method, we should submit it to the LLM track instead of the NLG track.
>
> **Q:** *Reason 2*
>
> **A:** We compare two major classes of baseline models, VAE-based transformer models and retrieval-generative models. Before we designed RegaVAE, DELLA was the latest VAE-based transformer model (as far as we know), and the rest of the models were its baselines. In the retrieval generation baseline model, they are all works with over 100 citations. Secondly, in 2022, Retro emerged as a state-of-the-art retrieval generation model. And FiD will occupy the first position on the KILT leaderboard all year round. So we consider our experiments to be adequate and comprehensive. In addition, under the same parameter scale, RegaVAE achieves the best performance among these models.
>
> **Q:** *Reason 3*
>
> **A:** The overall writing idea of this paper is as follows. In the field of retrieval generation, we have summarized two significant challenges, one is what information needs to be retrieved, and the other is how to integrate the retrieved information into the generation process. We then concluded two critical points based on previous research and experience. That is, valuable retrieval information should not only be related to the current source text but also consider the future target text, secondly, aggregation of latent variables derived from a compact latent space is more effective than exploiting explicit original text. Then we propose the RegaVAE framework, which simultaneously satisfies the two conditions of considering future information and compact latent space. Finally, we compared the retrieval-based baseline model with the VAE-based transformer baseline model to prove the superiority of RegaVAE. Evaluation metrics include text generation quality and hallucinations.
>
> **Q:** *Question A*
>
> **A:** Thank you for your reminder, due to misoperation, it should be $=$ but it was wrongly written as $>=$. We'll make changes. Nevertheless, it does not affect our derivation.
>
> **Q:** *Question B*
>
> **A:** In line 302 we point out that $p$ and $\hat{p}$ are Gaussian mixtures, so here $g_i$ and $\hat{g_i}$ are Gaussian distributions.
>
> **Q:** *Question C*
>
> **A:** We speculate that what you want to say here is $\hat{w_i}$. In lines 288~290, we use $\rm{softmax}$ to normalize $\hat{w_i}$. So $\sum \hat{w_i} = 1$ is correct.
>
> **Looking forward to further discussions with you.**

---

### Official Review · Reviewer_oBTQ · 2023-08-10

**Soundness:** 4

**Excitement:**

3: Ambivalent: It has merits (e.g., it reports state-of-the-art results, the idea is nice), but there are key weaknesses (e.g., it describes incremental work), and it can significantly benefit from another round of revision. However, I won't object to accepting it if my co-reviewers champion it.

**Missing References:**

None that I can identify.

**Paper Topic And Main Contributions:**

The authors propose a new retrieval augmented language model based on guassian mixture VAE. Retrieval augmented language models are proposed as a possible approach to resolve issues that stems from the scale of LLMs. The authors designed their model to implicitly aggregate future information in key, query, and values. The ability to aggregate future information implicitly on key and query is the defining property of their model. The authors detail their approach to the challenges of incorporating these features into the model. Their model is competitive against other baselines across different selected evaluation metrics on Yelp and Yahoo datasets.

**Questions For The Authors:**

Question A: Tab. 2 and 3, GPT2 is pre-trained only or further fine tuned?

Question B: What is the standard deviation of the scores reported in Table 5?

Question C: Line 460. 50 total samples for the 5 selected models? Meaning each model gets 10 questions?

Question D: I understand that there are considerations implementing this approach on LLMs. As the training comprises of additional training steps, how performant is it currently compared to fine-tuning GPT2? Will it be a challenge to extend this to LLMs?

Question E: I might have missed the information. In Figure 1, are all annotated VAEs in the diagram have the same weights? or are they the same model but different weights?

**Reasons To Accept:**

A1: The authors proposed a new retrieval augmented language model which combines VAE and text retrieval. It is competitive on widely-used text generation metrics, experiments for automated evaluations of generation quality and hallucinations.

A2: The authors conducted additional ablation studies and human evaluation in a study group comprising of 5 people.


**Reasons To Reject:**

No obvious reasons identified.

**Reproducibility:**

4: Could mostly reproduce the results, but there may be some variation because of sample variance or minor variations in their interpretation of the protocol or method.

**Reviewer Confidence:**

2: Willing to defend my evaluation, but it is fairly likely that I missed some details, didn't understand some central points, or can't be sure about the novelty of the work.

**Typos Grammar Style And Presentation Improvements:**

Include error bars in the evaluation results (if any), while some metrics are convincingly distant, there are metrics that are close (i.e. Dist2 in Tab3, S_BERT and S_QA in Tab. 4 and Tab. 5).

Sample questions and Instructions given to human evaluators can be included in the appendix.

Additional comparisons of complete generated text from various models can be included in the appendix.

---

> ### Author Rebuttal · Authors · 2023-08-28
>
> We thank the reviewer for the insightful comments. In the subsequent sections, we aim to meticulously address each of your concerns.
>
> **Q:** *Question A*
>
> **A:** In Tab. 2 and 3, the GPT2 model parameters are first initialized by the original GPT2 model on Huggingface, and then further fine-tuned.
>
> **Q:** *Question B*
>
> **A:** We calculated the standard deviation of the data in Tab. 5, as shown in the table below. These data will eventually be added to Tab. 5.
>
> Model  | Flu. | Coh. | Div. | Hal.
> :-: | :-: | :-: | :-: | :-:
> FiD | 0.28 | 0.26 | 0.35 | 0.28
> RETRO | 0.24 | 0.19 | 0.25 | 0.20
> RAG | 0.26 | 0.24 | 0.30 | 0.23
> DELLA | 0.23 | 0.20 | 0.27 | 0.30
> RegaVAE | 0.25 | 0.21 | 0.24 | 0.18
>
> **Q:** *Question C*
>
> **A:** Each model has 50 samples. To avoid misunderstanding, we will use a more precise expression.
>
> **Q:** *Question D*
>
> **A:** Compared to the fine-tuned GPT2, RegaVAE achieves clear improvements in all performances (Tab. 2, 3). Although an additional training step is included, we do not use additional training data. To ensure fairness, we keep the hyperparameters consistent. Although it will be difficult to extend RegaVAE to LLM, a feasible idea is to add a small number of fine-tunable parameters to the original LLM to implement the RegaVAE framework. This allows effective fine-tuning of parameters and reduces the cost of application on LLM.
>
> **Q:** *Question E*
>
> **A:** Fig. 1 shows the training process of RegaVAE, where all VAEs have the same weights.
>
> **Q:** *Typos Grammar Style And Presentation Improvements*
>
> **A:** For the main experiment, due to the limitation of computational cost (a model runs on a data set for an average of 3 days), we cannot determine the error bars during the rebuttal period, but we will supplement them before the final admission results come out. In addition, we will supplement relevant representative samples in the appendix.

---

### Official Review · Reviewer_MX59 · 2023-08-13

**Soundness:** 3

**Excitement:**

3: Ambivalent: It has merits (e.g., it reports state-of-the-art results, the idea is nice), but there are key weaknesses (e.g., it describes incremental work), and it can significantly benefit from another round of revision. However, I won't object to accepting it if my co-reviewers champion it.

**Missing References:**

Consider whether it'd be appropriate to cite previous work where a VAE is extended with a mixture distribution prior, such as a mixture of Gaussians (even though on a different domain/model architecture), for instance:

- "VAE with a VampPrior" (Tomczak and Welling, 2018)
- "Gaussian Mixture Variational Autoencoder with Contrastive Learning for Multi-Label Classification" (Bai, Kong and Gomes, 2022)


**Paper Topic And Main Contributions:**

This paper is mainly a NLP modeling + engineering experiment that addresses two problems in retrieval augmented language models (LMs):

1. selecting relevant information from the documents database -- the authors argue that all/most existing retrieval augmented LMs only use the input query ("current information") to retrieved documents/passages, leading to the information retrieved being similar to the context/query, and not necessarily containing relevant bits to continue the generation ("future information")
2. effectively combining retrieved information during generation -- the authors argue that explicit aggregation (concatenating retrieved documents to the source text) is limited by the length of the input and introduces noise, while implicit aggregation (using vectors or distributions) may be inefficient in irregular embedding spaces and retrieval vector not generalizable.

The main contributions of the paper are:

- proposing a novel retrieval-augmented LM based on a variational auto-encoder (VAE), which encodes the text corpus into a latent space that captures both current and future information;
- deriving an optimizable upper bound for the RegaVAE objective function;
- conducting experiments on three text generation datasets, and comparing RegaVAE performance with existing retrieval-augmented LMs and VAE-based LMs in terms of generation quality and hallucination reduction.

**Questions For The Authors:**

A. How is the model handling cases where the future information is not explicitly given or is ambiguous?

B. How did you handle the case when the retrieved documents are not relevant or helpful for the generation task? Did you have any mechanism to filter out or down-weight such documents?

C. Experiments: Table 3: Could you compute and report these scores for the retrieval model as well? What is a reason for Della performing better?

D. Experiments, human annotation: what were the definitions of Fluency, Consistency etc. given to/agreed by the human annotators?
Could you provide (in appendix?) representative examples of comparison between text generated with the various baseline models and the proposed model?

E. Have you done any hyperparameter optimization for the proposed model? (e.g. rank for the low-rank tensor product)
for the baseline models?
If so, how was the computing budget decided?

F. "due to limitation of computing resources, we cannot pre-train RegaVAE on large-scale corpus"
--> What were the compute resources/time/power used for training the model? what is the current energy requirement?
--> How much computing budget (and energy) do you estimate would be needed to train the proposed model on large scale compute and match the performance of large LMs?
I'm wondering whether, when training on much more data, the proposed model would need significanlty more compute to reach the level of current SOTA retrieval models.
(cf. last point on computing time/power in https://2023.eacl.org/ethics/faq/ and referenced impact tracker: https://arxiv.org/abs/2002.05651)

F. "bis" Also, can you list the computing resources/time/power used for training the baseline models and add it in the results tables, to allow for a more informed comparison between the models?

**Reasons To Accept:**

- this paper tackles 2 challenges (cf above) in retrieval-augmented text generation that (to my knowledge) have not been much studied in previous works.
- paper is very well-written and well-organized, and provides clear motivation, background, and some technical details for the proposed model, including a theoretical analysis of the optimizable upper bound for the model. It includes ablation studies and case studies.
- the proposed training procedure incorporates improvements/optimizations from prior work in this field.
- experiments were performed on three relevant datasets, with meaningful comparison against existing retrieval-augmented LMs and VAE-based LMs; they show that the performance of the proposed model is competitive in terms of generation quality and hallucination reduction. The evaluation metrics chosen are relevant. The paper also includes ablation studies and case studies to analyze the performance further.



**Reasons To Reject:**

- the paper may benefit from extending discussions on the potential drawbacks of the proposed model, e.g. computational cost (actual one, or expected cost to reach current large LMs performance), scalability, robustness.

- some missing experimental setup: hyperparameter search, computing budget, which implementation of the metrics was used
-- the code will surely give these experimentation details, when released -- it'd be great to shortly mention them in the paper itelf!

**Reproducibility:**

3: Could reproduce the results with some difficulty. The settings of parameters are underspecified or subjectively determined; the training/evaluation data are not widely available.

**Reviewer Confidence:**

3: Pretty sure, but there's a chance I missed something. Although I have a good feel for this area in general, I did not carefully check the paper's details, e.g., the math, experimental design, or novelty.

**Typos Grammar Style And Presentation Improvements:**

The paper is well-written, I did not notice any typo or grammatical error, apart from "RegaVAE outperforms the baseline models almost all --> "almost always" or "almost all the baseline models" (last sentence just before section 5 - Analysis)

+possible ~repetition in "Notably, […] there was a notable […]" (2nd sentence in subsection 5.2 Ablation experiment")

--

It may be difficult for a reader not familiar with the details of RETRO to quickly grasp what constitutes "future information" and how it is "extracted" or inferred from the text corpus.  Clarifying the definition, and being explicit (even if redundant) when mentioning what exactly is being considered as future information, may help them.

e.g. : "a natural idea is to divide a piece of text *in the document database/in the pool of retrievable documents* into two parts, […]"

--

Several claims could be better supported by references, for instance:

- intro: "[LMs] store a large amount of world knowledge as implicit parameters" (any study to cite here?)
- "transformers based on VAE usually have the problem of posterior collapse" (citing papers that show this?)
- "implicit aggregation methods can effectively filter noise compared to explicit aggregation methods" (is there some examples or analysis of this?)
- related work: on explicit aggregation: "it is easy to introduce noise" (which of the papers mentioned earlier in the paragraph develop this point? any example of such noise and its impact?)
- related work: pitfalls of implicit aggregations such as "implicit aggregation is inefficient in irregular embedding spaces" (any of the cited papers or another study to support, detail or illustrate this?)

---

> ### Author Rebuttal · Authors · 2023-08-28
>
> We thank the reviewer for the detailed and thorough review. We added the suggested experiments to the rebuttal revision. In the following, we seek to address each of your concerns.
>
> **Q:** *Computational cost mentioned in reason 1, and problem F*
>
> **A:** Taking the GPT2-based RegaVAE trained on the Yelp dataset as an example, it takes 2.5 days for full fine-tuning with a single A40-48G. The single A40 of the server we use has a maximum power of 300w. Based on this, it is inferred that the total energy consumed is 64,800 kJ. To extend RegaVAE to LLM, we will consider adding a small number of fine-tunable parameters to the original LLM to implement RegaVAE, such as RegaVAE based on LLama2. Compared with pretraining on large-scale data, this method is parameter efficient (see LoRA) and the computational cost is very low. Since the performance of some baseline models comes from the DELLA paper, the computational cost is not reported in it. Therefore, we will give the calculation cost of our own reproduced baseline models, where “h·single A40” represents the cost of training for 1h on a single A40. As shown in the following table:
> Model | Computational cost (h·single A40) |
> :-: | :-: |
> KNN-LM | 8|
> FiD | 66 |
> RAG | 58 |
> RETRO | 44 |
> RegaVAE | 60 |
>
> **Q:** *Scalability mentioned in reason 1*
>
> **A:** RegaVAE has a certain scalability. As mentioned before, LLMs-based RegaVAE can be realized by adding a small number of fine-tunable parameters to the original LLM. Furthermore, we have the flexibility to substitute various base models, such as Bart, T5, and others, in place of GPT2.
>
> **Q:** *Robustness mentioned in reason 1*
>
> **A:** We analyze the robustness of the model from the perspective of retrieving neighbors. Thanks to the implicit aggregation method, RegaVAE can filter out noise information well even with irrelevant documents (see Sec 5.1).
>
> **Q:** *Reason 2 and question E*
>
> **A:** All VAE-based models have the problem of posterior collapse, so we use strategies such as cyclic annealing, which is the same as previous work (DELLA, Embed, Memory, and Softmax). We did not perform hyperparameter optimization. Instead, to ensure fairness, we used the same hyperparameter settings as previous work (DELLA), including learning rate, Batch size, etc. Some experimental settings for training RegaVAE have been given in Sec 4.3. We will supplement the settings of other hyperparameters, and the corresponding implementation methods.
>
> **Q:** *Question A*
>
> **A:** The future information of a sentence is defined by us as its continuation. In the training phase, we construct a sentence (current information) and its continuation (future information) into sample-label pairs $(x, y)$ and feed them into RegaVAE to obtain a compact latent space. In the inference phase, we feed the test sample $x'$ into RegaVAE. At this time, the hidden variable $z$ sampled from the latent space contains the future information of $x'$. Therefore, there is no situation where future information is not expressly given or is unclear.
>
> **Q:** *Question B*
>
> **A:** We have considered this issue and conducted corresponding experiments (Sec. 5.1). We explored the impact of different numbers of neighbors on model performance on three datasets, and concluded that the implicit fusion method itself has a filtering mechanism for irrelevant neighbors (perhaps due to the attention mechanism in Transformer), the more neighbors retrieved within a certain range, the better the model performance. We can also set an explicit filtering mechanism, such as a similarity threshold to filter irrelevant documents. However, we think it is just a trick and has nothing to do with our research goals.
>
> **Q:** *Question C*
>
> **A:** WP is a story continuation data set. The average token length of each sample is 702, far exceeding 96 for Yelp and 79 for Yahoo. Computing the scores for retrieval augmentation models on top of this is therefore computationally expensive. Since we have demonstrated that RegaVAE outperforms these retrieval augmentation models on two datasets, Yelp and Yahoo, we do not consider reporting the scores of these retrieval models on the WP dataset. In addition, in Tab. 3, RegaVAE is better than DELLA in almost all metrics.
>
> **Q:** *Question D*
>
> **A:** Fluency refers to the fact that the text has good connections and readability. Coherence represents the layering and smooth cohesion of the text when expressing a specific theme. The example presented in Fig. 3 demonstrates this. We will show an example for each baseline model in the appendix for comparison.
>
> **Q:** *Missing References, Typos Grammar Style, And Presentation Improvements*
>
> **A:** Thank you for your reminder. These missing references are relevant to our theoretical framework and we cite them where appropriate. Relevant expressions will be corrected. We will add relevant references to support some of the claims.

---

### Meta-Review · Area_Chair_5SRC · 2023-09-20

**Recommendation:** 4

**Metareview:**

Proposes an approach to enhance the performance of Retrieval Augmented Language Models, as it relates to outdated information and hallucination. The idea is to make the retrieved information related to the source text but also take future target text into account. Based on this, the authors propose an approach based on aggregating latent variables from a latent space as opposed to using the original text and develop the RegaVAE framework, which is based on built using VAE. Theoretical analysis is conducted and an upper bound for the objective is developed. Experiments reveal significant improvement in text generation quality and hallucination reduction.

PLUS:

- addresses two key challenges in Retrieval Augmneted Language models: selecting relevant retireval info and effectively aggregating retrieved info during generation. Appears to be novel.
- theoretical analysis: upper bound on objective, which is optimizable.
- ablation study and case study.
- solid experimental study showing improvement over baselines on text generation quality and hallucination reduction.

MINUS:

- computational cost not analyzed.
- a few additional experiments could add value, e.g., hyperparameter search.

---

### Decision · Program_Chairs · 2023-10-07

**Decision:**

Accept-Findings

**Comment:**

Proposes an approach to enhance the performance of Retrieval Augmented Language Models, as it relates to outdated information and hallucination. The idea is to make the retrieved information related to the source text but also take future target text into account. Based on this, the authors propose an approach based on aggregating latent variables from a latent space as opposed to using the original text and develop the RegaVAE framework, which is based on built using VAE. Theoretical analysis is conducted and an upper bound for the objective is developed. Experiments reveal significant improvement in text generation quality and hallucination reduction.

PLUS:

- addresses two key challenges in Retrieval Augmneted Language models: selecting relevant retireval info and effectively aggregating retrieved info during generation. Appears to be novel.
- theoretical analysis: upper bound on objective, which is optimizable.
- ablation study and case study.
- solid experimental study showing improvement over baselines on text generation quality and hallucination reduction.

MINUS:

- computational cost not analyzed.
- a few additional experiments could add value, e.g., hyperparameter search.